# Dynamic Personalized Federated Learning with Adaptive Differential Privacy

Xiyuan Yang[1*]    Wenke Huang[1*]    Mang Ye[1†]

[1]National Engineering Research Center for Multimedia Software, Institute of Artificial Intelligence,
Hubei Key Laboratory of Multimedia and Network Communication Engineering,
School of Computer Science, Hubei Luojia Laboratory, Wuhan University, Wuhan, China.
{yangxiyuan,wenkehuang,yemang}@whu.edu.cn
https://github.com/xiyuanyang45/DynamicPFL

## Abstract

Personalized federated learning with differential privacy has been considered a feasible solution to address non-IID distribution of data and privacy leakage risks. However, current personalized federated learning methods suffer from inflexible personalization and convergence difficulties due to two main factors: 1) Firstly, we observe that the prevailing personalization methods mainly achieve this by personalizing a fixed portion of the model, which lacks flexibility. 2) Moreover, we further demonstrate that the default gradient calculation is sensitive to the widely-used clipping operations in differential privacy, resulting in difficulties in convergence. Considering that Fisher information values can serve as an effective measure for estimating the information content of parameters by reflecting the model sensitivity to parameters, we aim to leverage this property to address the aforementioned challenges. In this paper, we propose a novel federated learning method with **D**ynamic Fisher **P**ersonalization and **A**daptive Constraint (FedDPA) to handle these challenges. Firstly, by using layer-wise Fisher information to measure the information content of local parameters, we retain local parameters with high Fisher values during the personalization process, which are considered informative, simultaneously prevent these parameters from noise perturbation. Secondly, we introduce an adaptive approach by applying differential constraint strategies to personalized parameters and shared parameters identified in the previous for better convergence. Our method boosts performance through flexible personalization while mitigating the slow convergence caused by clipping operations. Experimental results on CIFAR-10, FEMNIST and SVHN dataset demonstrate the effectiveness of our approach in achieving better performance and robustness against clipping, under personalized federated learning with differential privacy.

## 1   Introduction

Federated learning (FL) [32] has shown great potential in facilitating collaborative machine learning across distributed devices. However, in real-world scenarios, local data normally present non-IID [40, 44, 43, 10, 16, 27, 26], (non-independent and identically distributed), posing data heterogeneity challenges [19, 22, 14, 9, 37] in FL. To address these challenges, various personalized federated learning (PFL) algorithms have emerged [39, 2, 30, 28], demonstrating significant effectiveness by incorporating personalized components that cater to the local data characteristics of each client. Despite their success, PFL faces severe privacy leakage issues, as client-local training data may be reconstructed through inference attacks [11, 33, 18, 42] during the collaborative updating phase. As a

---

[*]Equal contribution. † Corresponding author.

37th Conference on Neural Information Processing Systems (NeurIPS 2023).

result, it is essential to adopt user-level differential privacy (DP) techniques [12, 8, 41, 38] to provide more stringent privacy protection to PFL against potential privacy leaks, a.k.a, personalized federated learning with differential privacy.

Despite its pilot progress, personalized federated learning with differential privacy still faces numerous challenges. Most existing PFL approaches [30, 28, 3, 2] primarily rely on strong prior assumptions to perform parameter partitioning, personalizing a fixed portion of the model parameters and sharing the remaining ones. However, this approach lacks flexibility in parameter partitioning and cannot fully adapt to the diverse data characteristics of different clients, thereby affecting the personalized model performance. Besides, adding noise is an indispensable aspect in the differential privacy setting to ensure privacy protection in PFL. However, previous methods [8, 38, 3] directly add noise to all the network parameters, which unavoidably causes performance degradation. Considering that partial personalized parameters might not rely on the feedback from the server, this provides potential to reduce the noise impact. In light of these limitations, an essential issue that has long been overlooked is ❶ *how to develop a flexible personalization approach that enhances the learning capability of model while mitigating the impact of noise on the model.*

In order to achieve comprehensive user-level differential privacy, it is also crucial to implement a clipping operation targeting the L2 norm of each local update [12, 41]. These updates are sent to the server upon the completion of local training, ensuring that their magnitudes are bounded by the clipping bound. However, traditional optimization methods [12, 38] primarily focus on minimizing empirical risk, disregarding constraints on the L2 norms. Consequently, the local updates accumulated by gradients are susceptible to clipping, leading to distortion phenomena that can adversely affect model convergence. Existing methods for enhancing robustness [8, 23] against clipping often apply a uniform L2 regularization strength to all parameters. Such approaches neglect the inherent differences in parameter magnitudes, constraining the model learning capacity. This raises another crucial question: ❷ *how can we devise a dynamic constraint mechanism that maintains the learning capacity of the model as well as augments clipping robustness?*

In response to ❶, we propose an adaptive method that dynamically determines the personalized parameter portion for each client, while simultaneously mitigating the influence of noise. Our approach is inspired by the Fisher information matrix, which reflects the contribution of parameters to the curvature of the loss function by computing the square of the first-order derivatives of log-likelihood function. This mathematical process can also be intuitively understood as the information content carried by the parameters. Following this concept, we further introduce layer-wise Fisher information to measure the information content of client parameters. Prior to client training, we retain essential parameters as personalized components, implementing personalization by preventing them from being overwritten by the corresponding noisy global parameters. By employing this adaptive personalization strategy, our method ensures that locally retained parameters remain unaffected by the noise introduced for privacy protection, effectively overcoming the challenges associated with inflexible personalization and the impact of noise.

To tackle the issue of local update robustness to ❷, we introduce an adaptive method that determines different constraint strengths for different parameters. Taking into account the inherent differences in parameter magnitudes, we only apply L2 norm regularization constraints to the updates of personalized parameters selected on account of layer-wise Fisher information, which preserves more unique knowledge of local data, to maintain local knowledge effectively. While for the shared parameter updates, we ensure that their L2 norm values approach the clipping bound through a bounded regularization term, effectively mitigating the impact of clipping operation. By adopting these strategies, we enhance the robustness to clipping while addressing the limitations of uniform regularization successfully.

In this work, we propose a novel personalized federated learning method under differential privacy named FedDPA via **D**ynamic Fisher **P**ersonalization (DFP) and **A**daptive Constraint (AC). Our main contributions are summarized as follows:

- We propose a dynamic personalization strategy using layer-wise Fisher information values, enabling flexible personalization and reduced noise impact.

- We present an adaptive method for enhancing clipping robustness by adjusting constraint strengths for different parameters, balancing client-specific features and global knowledge while minimizing clipping effects.

- We conduct extensive experiments under multiple tasks and settings. Accompanied with a set of ablative studies, promising results validate the efficacy of our proposed methods and indispensability of each module.

## 2   Related Works

### 2.1   Personalized Federated Learning

Federated Learning is a distributed machine learning approach that allows data to remain on local devices while training a global model, with the FedAvg algorithm [32] as its most representative example. Despite its revolutionary approach, the inherent non-Independent and Identically Distributed (non-IID) nature of local data [19, 22, 40, 44, 17, 13, 15] poses significant challenges, thus leading to the advent of personalized federated learning (PFL). In PFL, a portion of the model is personalized to learn client-specific knowledge, while the remaining parts aim to capture the global patterns across all clients. The mainstream PFL methods include LG-FedAvg [28], FedPer [2], PPSGD [3], and FULR [5]. The LG-FedAvg [28] and FULR [5] methods both leverage a fixed local parameter to extract a representation of the local data, thereby achieving personalization, but their static partitioning approach lacks flexibility in adapting to diverse data characteristics. Besides, the FedPer [2], FedBABU [35] and PPSGD [3] algorithms achieve personalization by retaining specific layers locally, which can better cater to local data characteristics, but suffer from potential inflexibility in model personalization. Additionally, another work [36] introduces FedSim and FedAlt, which analyze different update paradigms of local and global parameters, contributing to the understanding of the trade-offs between local data specificity and global data commonality. These approaches, while effective to a certain extent, still bear limitations in their inflexible of personalization methods and the potential performance degradation due to the direct training with noisy global parameters under DP mechanism. We introduce an innovative approach to personalized federated learning. Our method, based on layer-wise Fisher information values, allows for dynamic personalization that surpasses the inflexibility of fixed parameter partitioning while mitigating the noise impact.

### 2.2   Differential Privacy

Differential Privacy (DP) provides substantial privacy in federated learning [12, 8, 41, 38, 25, 24], by minimizing the impact of individual data changes on computation outputs. This privacy level is quantified via $(\varepsilon, \delta)$-differential privacy. Smaller $\varepsilon$ and $\delta$ offer stronger privacy but introduce more noise into the learning algorithm, which could degrade performance.

In federated learning, user-level DP is achieved by a two-step process intrinsic to the DP mechanism: adding noise and clipping local updates before server transmission. The noise scale is calibrated according to the function sensitivity. Clipping, which bounds the impact of local update, further reinforces privacy. Although these essential steps ensure substantial privacy, they present challenges, such as performance degradation and slower convergence, due to the unavoidable noise and clipping operation. While user-level DP ensures robust privacy, it introduces challenges such as performance degradation from added noise and slower convergence due to clipping. These challenges have been addressed by several methods, including LUS [8], BLUR [8], DP-FedSAM [38], and PPSGD [3]. LUS and BLUR employ sparsification and unified regularization techniques to mitigate the effects of added noise and enhance model convergence. DP-FedSAM [38] utilizes the SAM optimizer to bolster parameter robustness against noise, aiming to identify more stable convergence points. Besides, PPSGD [3] capitalizes on personalization to improve performance while maintaining privacy. Despite their promising solutions, these methods don't sufficiently address the issues caused by noise and clipping in differential privacy. They failed to find the essence of the problem of noise perturbation, all parameters will be affected by noise during training, leading to the degradation of model performance. Furthermore, their uniform handling of clipping constraints fails to adapt to the dynamic nature of local updates, which further hinders model learning and convergence. To tackle these challenges, we propose a novel method that leverages layer-wise Fisher information to dynamically personalize and protect parameters of high information content from noise impact. Simultaneously, we introduce an adaptive regularization strategy to enforce differential constraints on personalized and shared parameters, enhancing the model learning capacity and robustness to clipping.

## 2.3 Fisher Information Matrix

The Fisher Information Matrix [29, 1], a pivotal concept in statistical estimation theory, encapsulates the information an unknown parameter possesses about a random distribution. In the realm of deep learning, FIM has been employed to study the curvature of loss functions, guide optimization and evaluate the information content of parameters [31, 4, 20]. For instance, the Kronecker-Factored Approximate Curvature (K-FAC) method [31] utilizes a Kronecker product approximation to the FIM for more efficient natural gradient computations. The layer-wise relevance propagation method [4] uses the diagonal of the FIM to quantify the importance of features, thereby enhancing model interpretability. The Elastic Weight Consolidation algorithm [20, 7] employs FIM to guard important parameters during the learning of new tasks, mitigating catastrophic forgetting. All these methods utilize the diagonal of the FIM for approximation, reducing computational complexity and facilitating more efficient learning processes. Although these methodologies have made significant strides, their application in personalized federated learning with differential privacy constraints, remains largely unexplored. These methods are not directly applicable to personalized federated learning, as they don't account for the unique challenges posed by non-IID data distribution and privacy constraints. Our novel approach integrates layer-wise Fisher information into personalized federated learning, leveraging it to dynamically determine the personalization parameters, thus protecting informative ones from noise perturbation induced by differential privacy. This strategy effectively minimizes the impact of noise, enhancing model performance in personalization. Additionally, we introduce an adaptive method that dynamically adjusts L2 norm constraints based on their fisher information value magnitudes. This method enhances the robustness to clipping operations, offering a more flexible solution to balance between model learning and privacy protection under the PFL with differential privacy.

# 3 Methodology

## 3.1 Preliminary

**Personalized Federated Learning.** Personalized Federated Learning (PFL) extends Federated Learning (FL) by addressing the non-IID nature of real-world data distributions. In PFL, parameter vector of client model is decomposed into a local part, $u$, that caters to the individual data peculiarities, and a global part, $v$, which is universally shared.

Assume we have $M$ clients, Each client possesses a unique private dataset, denoted by $D_m = (x_i, y_i)_{i=1}^{N_m}$, where $N_m$ signifies the size of the local dataset of client $m$. The parameter vector $w \in \mathbb{R}^{d_w}$ of model of client $i$ is separated into local and global parts, forming $w_i = (v, u_i)$, by personalization techniques.

The optimization problem in PFL is defined as follows:

$$\min_{v, u_{1:m}} \{ f(v, u_{1:m}) := \frac{1}{m} \sum_{i=1}^{M} f_i(v, u_i) \}, \tag{1}$$

where $u_{1:m}$ represents $(u_1, \ldots, u_m) \in (\mathbb{R}^{d_u})^M$ and the $f_i(v, u_i) := \mathbb{E}_{D_i \sim P_i}[f_i(v, u_i, D_i)]$ is the expected risk of client $i$, where $i = 1, ..., N_m$. PFL use a iteration of two steps to solve this problem:

- **Local Update:** During the local update phase, each client $i$ adopts the latest global parameter $v^t$ while retaining its local $u_i^t$ from previous round to initialize the model $w_i^t = (v^t, u_i^t)$, and then performs $E_{local}$ iterations of local updates to obtain new parameters $w_i^{t+1} = (v_i^{t+1}, u_i^{t+1})$. Subsequently, they compute the local update $\Delta v_i^{t+1}$ by calculating the difference between $v_i^{t+1}$ and $v_i^t$.

- **Collaborative Update:** In the collaborative update phase, all clients send their local updates $\Delta v_i^{t+1}$ to the server. The server then averages these parameters to update the parameters $v^{t+1} = v^t + \frac{1}{M} \sum_{i=1}^{M} \Delta v_i^{t+1}$, which are then distributed back to all clients for the next round of local updates. In our methods, since the shared part of parameters is selected dynamically and cannot be previously determined , all clients send their whole updates $\Delta w_{t+1}^t$ to server.

**User-Level Differential Privacy.** In the context of Personalized Federated Learning (PFL), it is essential to employ differential privacy techinque to prevent potential privacy leakage. One strong privacy-preserving mechanism used in this setting is Differential Privacy (DP).

Differential Privacy (DP) [12, 41] is a formal privacy framework that provides theoretical guarantees against the identification of private data in a dataset. In the DP framework, $\epsilon$ is known as the privacy budget, which quantifies the privacy protection level, and $\delta$ is the probability of the privacy guarantee being violated. A randomized algorithm $M$ satisfies $(\epsilon, \delta)$-DP if for any pair of adjacent datasets $D$ and $D'$ differing by only one record, and for any subset of outputs $S$ in the range of $M$, it holds that:

$$\Pr[M(D) \in S] \leq e^\epsilon \Pr[M(D') \in S] + \delta \tag{2}$$

This property implies that the probability distribution of the output of $M$ changes only slightly when a single record in the input dataset is modified, thus protecting individual privacy.

User-Level Differential Privacy (DP) is extensively used as an application of DP mechanism in the context of personalized federated scenarios. User-Level DP refers to the privacy guarantee for the participation information of a single client in the learning process. Based on the aforementioned theory, user-level DP makes the global model updates indistinguishable whether a particular client participates in the learning or not. This is achieved by implementing two crucial operations: clipping and noise addition.

Clipping is done to control the maximum contribution of a single client to the global update, ensuring that the DP property holds even in the presence of outliers. This is crucial for maintaining the privacy of individuals in the learning process. Besides, noise addition (typically sampled from a zero-mean Gaussian distribution) is performed to satisfy the randomness requirement of DP, making it harder to infer specific information about any individual client. The combined effect of these two operations is expressed as:

$$\Delta w_C = \Delta w / max(1, \frac{||\Delta w||_2}{C}), \quad \Delta w_{C,N} = \Delta w_C + \mathcal{N}(0, C^2\sigma^2/|M_t|), \tag{3}$$

where $\Delta w$ represents the original local update from the client, $|M_t|$ represents the available client in round $t$, $C$ is the clipping threshold that controls the maximum contribution of a client to the global update, $\Delta w_C$ is the local update after clipping, $\Delta w_{C,N}$ is the final local update after clipping and adding noise, $\mathcal{N}$ is the Gaussian noise added to ensure DP, and $\sigma$ is a noise multiplier computed by privacy accountant and composition mechanism with respect to $\epsilon$ and $\delta$.

## 3.2 Dynamic Personalization Strategy

**Motivation of Fisher Information.** The motivation to use Fisher Information stems from its ability to quantify the amount of information that parameters can provide. Parameters that are more informative can represent the knowledge more effectively, playing a crucial role in the model prediction. Therefore, under the same noise perturbation, if the informative parameters are affected, it will lead to a more severe degradation of model performance. In typical personalization work, at the beginning of the local update phase, clients directly place the global parameters in fixed positions into the network for training. This inflexible personalization approach doesn't consider the impact of noise on different parameters, which results in the model being more susceptible to noise interference. Therefore, we need to avoid noise affecting informative parameters. In light of this, our approach considers the hierarchical structure of neural networks and introduces layer-wise Fisher information. We measure layer-wise of each parameter Fisher information, which reflects the sensitivity of the model to the parameter changes, to optimize the model by preventing them from noise.

**Construction of Dynamic Fisher Personalization.** As discussed in ❶, the current DP-PFL methods struggle with inflexible personalization due to the fixed portion of the model used for personalization. To address this issue, we introduce a dynamic personalization strategy that leverages the Fisher Information to enhance the adaptability of the model to the data distribution of individual clients as follows. We start at the beginning of local update phase of client $i$ at global epoch $t$, with its private dataset $D_i$ and parameters $w_i^{t-1} = (v^{t-1}, u_i^{t-1})$ remain from last epoch. We can get the empirical fisher value vector $F_i \in \mathbb{R}^{d_w}$ where $d_w$ represents the num of parameters of $w$ as a good approximation of the diagonal of true fisher value for each parameter indexed by $j$ in $w_i^{t-1}$ as :

$$F(w_{ij}) = \left(\frac{\partial \log L(w_i, D_i)}{\partial w_{ij}}\right)^2, \tag{4}$$

where the $\log L(w_i, D_i)$ represents log-likelihood function of $w_i$. Then, the layer-wise fisher value $\hat{F}_k$ for layer $k$ by a layer-wise normalization for each parameters indexed by $j$ in layer $k$ as :

$$\hat{F}_{k,j} = \frac{F_{k,j} - \min\{F_{k,j}\}}{\max\{F_{k,j}\} - \min\{F_{k,j}\}}. \tag{5}$$

Given the layer-wise Fisher values for each parameter, we then generate two binary masks, $M_1$ and $M_2$, for each parameter to conduct our dynamic parameter selection. In each mask, a parameter entry is set to 1 if its corresponding Fisher value is greater than or equal to a threshold $\tau$, otherwise, it is set to 0. These masks are defined as follows for each parameter $j$:

$$M_1[j] = \begin{cases} 1, & \text{if } \hat{F}_{ij} \geq \tau \\ 0, & \text{otherwise} \end{cases} \text{ and } M_2[j] = \begin{cases} 0, & \text{if } \hat{F}_{ij} \geq \tau \\ 1, & \text{otherwise} \end{cases}, \tag{6}$$

where $M_1$ and $M_2$ are the entries in the first and second masks, respectively.

We then perform a Hadamard product (element-wise multiplication) between these masks and the parameters to select the appropriate parameters for personalization. Specifically, parameters with a larger Fisher value, which correspond to 1's in $M_1$, are retained from the previous epoch, while the remaining parameters, which correspond to 1's in $M_2$, are replaced by the global parameters. This operation can be expressed as follows:

$$w_i^t = M_1 \odot w_i^{t-1} + M_2 \odot w^{t-1}, \tag{7}$$

where $\odot$ denotes the Hadamard product and $w^{t-1}$ is the global parameters downloaded from server. This approach ensures that more informative parameters (with larger Fisher values) are retained for personalized learning, while less informative ones are updated with the global parameters, mitigating the influence of noise.

**Comparison with Analogous Methods.** PPSGD [3] employs a fixed layer personalization approach with an additive model to achieve personalization under differential privacy (DP). Meanwhile, the LUS mechanism [8] utilizes a mask during their sparsification process after local training. Although these methods have their merits, they are fundamentally different from our proposed strategy. In our work, we advocate for a dynamic personalization approach using a mask, but crucially, this is applied prior to the training process. Our aim is to provide a more flexible and adaptive personalization strategy that can better cope with changes in the data distribution, noise level, and other dynamic factors affecting the model performance. Furthermore, we focus on preserving the most informative parameters, those that significantly contribute to the model performance, while allowing less informative ones to be updated with global parameters, thereby reducing the impact of noise. This dynamic, pre-training personalization strategy provides a unique advantage over the static, post-training personalization of the aforementioned methods.

### 3.3 Adaptive Constraint

**Motivation for Adaptive Constraint.** The challenge of gradient distortion and convergence difficulties in DP-PFL arises from the clipping operation applied to the L2 norm of each local update. Traditional methods[8, 23] often apply a uniform regularization strength to all parameters, neglecting the inherent differences among parameter magnitudes. To address this issue, we propose an adaptive approach that determines different constraint strengths for different parameters, considering the inherent magnitude of each parameter. This aims to maintain the learning capacity of the model while also enhancing its robustness against clipping.

**Construction of Adaptive Constraint.** As discussed in ❷, the problem of gradient distortion and convergence difficulties in DP-PFL arises from the clipping operation applied to the L2 norm of each local update, our method is as follows. At the start of the local update phase, each client $i$ at global epoch $t$ has parameters $w_i^{t-1} = (v^{t-1}, u_i^{t-1})$. These parameters are divided into two sets: shared parameters $v_i^{t-1}$ and personalized parameters $u_i^{t-1}$, identified by the previously introduced dynamic personalization strategy. We propose two different constraints for the local updates of these two sets of parameters.

For the personalized parameters, which contain unique knowledge from the local data, our aim is to retain this local knowledge as much as possible. We therefore introduce a regularization term to loss

function that constrains the L2 norm of the local update of the personalized parameters. The local update of personalized parameters $u_i^t$ is expressed as:

$$u_i^t = u_i^{t-1} - \eta \nabla_u \mathcal{L}_1(v_i^{t-1}, u_i^{t-1}, D_i), \tag{8}$$

where $\eta$ is the learning rate, and $\nabla \mathcal{L}_1$ is the gradient of the first loss function. The first loss function, $\mathcal{L}_1$, is calculated as the sum of the cross-entropy loss and the L2 norm of the difference between the current and the previous personalized parameters:

$$\mathcal{L}_1 = -\frac{1}{n} \sum_{j=1}^n [y_j \log(\hat{y}_j) + (1 - y_j) \log(1 - \hat{y}_j)] + \frac{\lambda_1}{2} ||u_i - u_i^t||_2, \tag{9}$$

where $n$ is the number of instances in the dataset $D_i$, $y_j$ is the true label of instance $j$, $\lambda_1$ is a hyper-parameter and $\hat{y}_j$ is the predicted label of instance $j$.

For the shared parameters, we strive to ensure that their L2 norm values closely approach the clipping bound. By adopting such a strategy, we anticipate reducing the influence that the clipping operation exerts on the local update. We therefore introduce a bounded regularization term to the loss function of the shared parameters. The local update of shared parameters $v_i^t$ is expressed as:

$$v_i^t = v_i^{t-1} - \eta \nabla_v \mathcal{L}_2(v_i^{t-1}, u_i^t, D_i), \tag{10}$$

where $\nabla \mathcal{L}_2$ is the gradient of the second loss function. The second loss function, $\mathcal{L}_2$, is calculated as the sum of the cross-entropy loss and the absolute difference between the L2 norm of the difference between the current and previous shared parameters and the clipping bound:

$$\mathcal{L}_2 = -\frac{1}{n} \sum_{j=1}^n [y_j \log(\hat{y}_j) + (1 - y_j) \log(1 - \hat{y}_j)] + \frac{\lambda_2}{2} \big| ||v_i - v_i^t||_2 - C \big|_2. \tag{11}$$

Here, $n$ is the number of instances in the dataset $D_i$, $y_j$ is the true label of instance $j$, $\hat{y}_j$ is the predicted label of instance $j$, $\lambda 2$ is also a hyper-parameter and $C$ is the clipping bound.

By applying these adaptive constraints, our method effectively mitigates the impact of the clipping operation and boosts the performance of FedDPA. The two loss functions, $\mathcal{L}_1$ and $\mathcal{L}_2$, cater to the unique properties of personalized and shared parameters, respectively. $\mathcal{L}_1$ helps retain as much local knowledge as possible, overcoming the slow convergence issue. Additionally, $\mathcal{L}_2$ ensures the shared parameters updates align with the clipping bound, thereby improving the model robustness against clipping. In both cases, the regularization strengths $\lambda_1$ and $\lambda_2$ are determined adaptively, taking into account the inherent magnitude of each parameter. This novel consideration distinguishes our method from traditional approaches and contributes to its superior performance.

### 3.4 Discussion and Limitation

Our analysis of extra computation cost is as follows: Assuming a total of $E_g$ Global Epochs, $M$ clients, and $P$ parameters for each model, during each client update process, the client undergoes $S$ steps of parameter updates in training. In each round, our method introduces computation costs as follows: 1) Computing the Fisher information value, 2) Construction of the mask and 3) Calculating the initial parameters using masks. In these steps, as each step is parameter-specific, our method brings an additional computational cost of $\mathcal{O}(P)$. Regarding that we have to carry out these operations in each global epoch, we introduce a computational cost of $\mathcal{O}(E_g \cdot M \cdot P)$ in total.

Despite our extra computational cost, it is noteworthy that our approach has a relatively small overhead compared to the overall training process. This is because, the model training still has a considerable computational cost proportional to $P$, and all the training process would introduce $\mathcal{O}(E_g \cdot M \cdot P \cdot S)$. Generally, in a training process, the value of $S$ is in the tens or hundreds, so the computation cost introduced by our method will be an order of magnitude or more less than the training expense. Extensive experiments have verified the consistent improvement over the counterparts.

Besides the core functionality of our model, it also shows potential for applications in other fields, such as feature selection in high-dimensional datasets, improving model robustness by focusing on critical parameters, and enhancing privacy protection by selective parameter sharing. Regarding the limitation of model homogeneity, our method assumes all clients share the same model structure, which may not always be the case in real-world scenarios. Heterogeneity in model structures across

**Algorithm 1:** The Proposed Method: FedDPA

---

**Input:** Global epochs $E_g$, local epochs $E_l$, participants number in the $t^{th}$ epoch $m^t$, private data of the $i^{th}$ client $D_i = (X_i, Y_i)$, global model parameters $w$ and client local parameters $w_i$, hyper-parameter $\lambda_1, \lambda2, \tau$, learning rate $\eta$

**Local Update :**
**for** $i = 1, 2, ..., m^t$ **do**
    Receive $w^{t-1}$ from Server
    $\hat{F}_i \leftarrow (D_i, w_i^{t-1})$ by Eq. (4) and Eq. (5)
    $M_{1i}$ and $M_{2i} \leftarrow (\hat{F}_i, \tau)$ by Eq. (6)
    $w_i^t = (v^{t-1}, u_i^{t-1}) \leftarrow M_{1i} \odot w_i^{t-1} + M_{2i} \odot w^{t-1}$ using Eq. (7)
    **for** $e = 1, 2, ..., E_l$ **do**
        $u_i^t \leftarrow u_i^{t-1} - \eta \nabla_u \mathcal{L}_1(v^{t-1}, u_i^{t-1})$ by $\mathcal{L}_1$ from (9)
        $v_i^t \leftarrow v^{t-1} - \eta \nabla_v \mathcal{L}_2(v^{t-1}, u_i^t)$ by $\mathcal{L}_2$ from (11)
    **end**
    $\Delta w_i^t \leftarrow (v_i^t, u_i^t) - (v^{t-1}, u_i^{t-1})$
    $\Delta w_i^t \leftarrow \Delta w_i^t / \max(1, \frac{||\Delta w_i^t||_2}{C}) + \mathcal{N}(0, C^2\sigma^2/|m^t|)$ to ensure DP by Eq. (3)
**end**
**Server Execute :**
$\Delta w_i^t \leftarrow$ **Local Update**$(w^{t-1})$
**for** $t = 1, 2, ..., E_g$ **do**
    $w^t \leftarrow w^{t-1} + \frac{1}{m^t} \sum_{i=1}^{m^t} \Delta w_i^t$
    **for** $i = 1, 2, ..., m^t$ **do**
        Send $w^t$ to $i^{th}$ participant
    **end**
**end**

---

clients can potentially limit the utility of our dynamic personalization strategy and adaptive constraints, differences in model could lead to incompatibilities, which is an area where further research is needed to improve the versatility of our method. It should be noted that this limitation is not unique to our approach but is also shared by most of the aforementioned methods [8, 41, 38, 3] .

# 4 Experiments

We evaluate the performance of the proposed method and comparison methods through comprehensive experiments on two distinct datasets under various conditions. Specifically, we test the accuracy on the FEMNIST and SVHN dataset across different privacy budgets denoted by $\epsilon$, and for the CIFAR-10, we adjust the $\alpha$ parameter to simulate varying degrees of non-IID distribution and assess accuracy accordingly. This wide-ranging experimental setup reflects practical scenarios and ensures a thorough assessment of our method's robustness and adaptability.

**Data and Model.** Our method is evaluated on two classification tasks, FEMNIST [6], CIFAR-10 [21] and SVHN [34], embodying real-world non-IID and privacy-constrained scenarios. The FEMNIST is a 62-class version of MNIST, CIFAR-10 is a widely-used image classification dataset, and the SVHN dataset it a digits classification dataset which is collected from street view. We use a simple CNN model with 2 convolution layers and 2 fully connected layers for FEMNIST and a deeper model with 3 convolution layers and 3 fully connected layers for CIFAR-10.

**Comparison Methods.** We compare the performance of our proposed method with several state-of-the-art methods in differential privacy-enabled federated learning, including DP-FedAvg [12], PPSGD [3], DP-FedAvg with LUS and BLUR [8], and DP-FedSAM [38].

**Evaluation Metrics.** The primary metric for evaluation is the average accuracy, calculated uniquely for each client. Each client has their own personalized model trained on their individual dataset, and accuracy is measured on these respective datasets. The average of these individual accuracies across all clients is then calculated to yield the final metric. For the FEMNIST and SVHN dataset, we evaluate this average accuracy under varying privacy budgets, underlining the advantage of our method in mitigating noise impact. For the CIFAR-10 dataset, we evaluate average accuracy

under different degrees of non-IID data partition, demonstrating the effectiveness of our dynamic personalization strategy in handling non-IID data.

**Implementation Details.** For all dataset FEMNIST, CIFAR-10 and SVHN, we set the learning rate to 1e-3 and optimize hyperparameters $\tau$, $\lambda_1$, and $\lambda_2$ through grid search in {0.05, 0.1, 0.3, 0.5}. We use the Rényi Differential Privacy (RDP) algorithm provided by Opacus as our privacy accountant. The value of $\delta$, in $(\varepsilon, \delta)$-differential privacy, is set to the reciprocal of the number of clients (*i.e.*, 1/M). We use global epochs of 30 and 40, local epochs of 3 and 4, and batch sizes of 16 and 64 for FEMNIST and CIFAR-10, respectively. For CIFAR-10, we partition the dataset into 10 subsets via Dirichlet distribution and use a fixed noise multiplier to ensure differential privacy. For FEMNIST, we selected the top 50 clients from the FEMNIST [6] division for training. All experiments were implemented in Python with PyTorch on an NVIDIA 3090 GPU.

Table 1: Comparison of Model Performance under Different Conditions. Refer Sec. 4.1

| Methods | FEMNIST | | | | SVHN | | | |
|---|---|---|---|---|---|---|---|---|
| | $\epsilon=2$ | $\epsilon=4$ | $\epsilon=8$ | $\epsilon=16$ | $\epsilon=2$ | $\epsilon=4$ | $\epsilon=6$ | $\epsilon=8$ |
| DP-FedAvg [12] | 71.92 | 72.80 | 73.26 | 75.02 | 53.91 | 54.76 | 56.33 | 57.65 |
| BLUR+LUS [8] | 72.80 | 74.23 | 74.58 | 75.16 | 53.76 | 57.52 | 56.90 | 58.57 |
| PPSGD [3] | 68.20 | 69.60 | 71.03 | 71.93 | 55.89 | 56.15 | 56.74 | 59.37 |
| DP-FedSAM [38] | 73.13 | 73.94 | 74.54 | 74.66 | 53.04 | 52.84 | 54.03 | 56.08 |
| FedDPA(Ours) | **74.46**↑1.33 | **77.27**↑3.04 | **77.42**↑2.84 | **76.99**↑1.83 | **58.78**↑2.89 | **62.63**↑5.11 | **63.66**↑6.76 | **64.57**↑5.29 |

Table 2: Performance Comparison on CIFAR-10. Refer Sec. 4.1

| Methods | $\alpha=1$ | $\alpha=10$ | $\alpha=100$ | IID |
|---|---|---|---|---|
| DP-FedAvg [12] | 42.61 | 58.89 | 60.03 | 60.97 |
| BLUR+LUS [8] | 48.97 | 58.70 | 60.50 | 61.12 |
| PPSGD [3] | 44.04 | 57.31 | 60.20 | 60.02 |
| DP-FedSAM [38] | 45.07 | 56.15 | 58.87 | 60.77 |
| FedDPA(Ours) | **49.75**↑0.78 | **59.48**↑0.59 | **60.68**↑0.18 | **61.55**↑0.43 |

## 4.1 Comparison with State-Of-the-Art Methods

We provide comparison results with SOTA methods on two image classification tasks [21, 6].

**Analysis Under Different Privacy Budget.** As Tab.1 shows, our method maintains high accuracy across various privacy budgets on the FEMNIST dataset, underlining its robustness and superior privacy-utility trade-off. The convergence plots Fig.1a and Fig.1b support our method's efficiency, demonstrating faster convergence and more effective loss minimization.

**Analysis Under Different Degrees of Non-IID Data.** Our method shows strong performance across varying non-IID data partitions on the CIFAR-10 dataset, as seen on the right side of Tab.2. It effectively handles non-IID data challenges via our flexible personalization method.

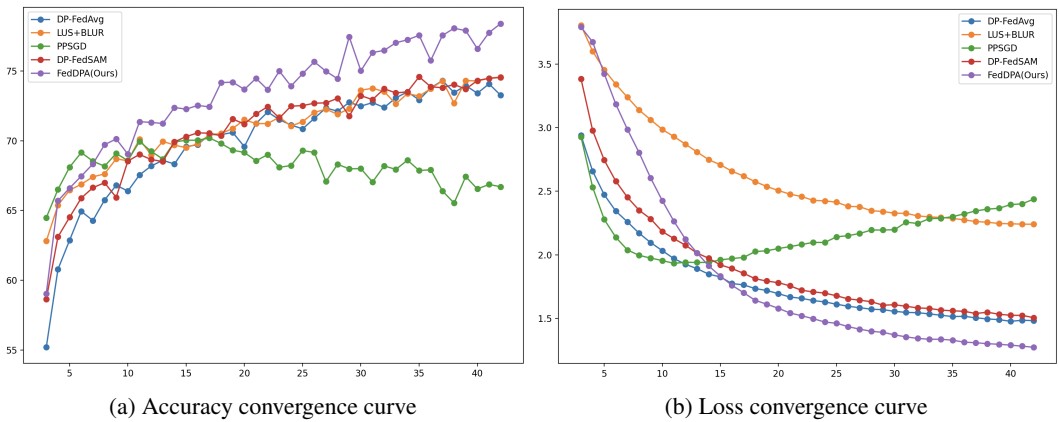

(a) Accuracy convergence curve      (b) Loss convergence curve

Figure 1: Comparison of Model Performance and Optimization Objective Value. See details in 4.1

## 4.2 Diagnostic Experiments

To highlight the contribution of each component in our method towards the overall performance, we conduct a series of ablation experiments. Our proposed method comprises two integral components.

**Dynamic Fisher Personalization (DFP).** The DFP, as described in Sec. 3.2, contributes to the improvement of accuracy across various privacy budgets. When DFP is employed (second row in the table), there is a noticeable increase in accuracy for all privacy budgets compared to the base model (first row), validating the effectiveness of DFP in enhancing model performance.

**Adaptive Constraint (AC).** The AC strategy, as introduced in Sec. 3.3, is dependent on DFP for its operation. Therefore, there is no experiment featuring only the AC without the DFP. However, when both DFP and AC are incorporated (third row in the table), the model's accuracy further improves across all privacy budgets. This demonstrates that the AC strategy enhances the utility of DFP, contributing to an overall superior privacy-utility trade-off.

Table 3: Ablation Study with Different Privacy Budgets. Refer Sec. 4.2

| Components | | Accuracy on clients dataset | | | |
|---|---|---|---|---|---|
| DFP | AC | $\epsilon = 2$ | $\epsilon = 4$ | $\epsilon = 8$ | $\epsilon = 16$ |
| | | 71.92 | 72.80 | 73.26 | 75.02 |
| ✓ | | 72.18 | 74.56 | 74.38 | 75.33 |
| ✓ | ✓ | **74.46**↑2.28 | **77.27**↑2.71 | **77.42**↑2.93 | **76.99**↑1.66 |

The results in Table 3 clearly illustrate that both DFP and AC contribute significantly to the performance of the model under various privacy budgets. The combined use of both components provides the best results, reinforcing the effectiveness of our proposed method.

## 4.3 Hyperparameter Study

We vary the hyperparameters including $\lambda$ and $\tau$ in our proposed methods to demonstrate how out model retains or change personalized portions in DFP and the sensitivity of constraint strength in AC.

Table 4: Ablation Study on $\tau$. Refer Sec. 4.3

| $\tau$ | 0 | 0.1 | 0.2 | 0.3 | 0.4 | 0.5 | 0.6 | 0.7 | 0.8 | 0.9 | 1 |
|---|---|---|---|---|---|---|---|---|---|---|---|
| Proportion | 100% | 11.1% | 4.5% | 1.45% | 0.97% | 0.46% | 0.15% | 0.1% | 0.1% | 0.05% | 0% |
| Accuracy | 39.72 | 47.97 | **49.28** | 48.89 | 48.35 | 48.03 | 48.28 | 48.14 | 48.35 | 47.74 | 48.29 |

Table 5: Ablation Study on $\lambda_1$. Refer Sec.4.3

| $\lambda_1$ | 0 | 0.05 | 0.1 | 0.3 | 0.5 |
|---|---|---|---|---|---|
| Accuracy | 60.82 | **62.64** | 59.66 | 59.28 | 59.57 |

Table 6: Ablation Study on $\lambda_2$. Refer Sec. 4.3

| $\lambda_2$ | 0 | 0.05 | 0.1 | 0.3 | 0.5 |
|---|---|---|---|---|---|
| Accuracy | 58.34 | 59.76 | **61.01** | 47.61 | 45.66 |

For $\tau$, we observe that our proposed method is relatively stable, verifying the robustness of our proposed solution. For $\lambda$, we achieve the best performance when $\lambda_1$ is around $0.05$ and $\lambda_2$ is around $0.1$. We apply this across multiple different datasets, and it results in a consistent good performance.

## 5 Conclusion

In this paper, we delve into the challenges of inflexible personalization and convergence difficulties under differential privacy in personalized federated learning. We introduce FedDPA with two innovative components, Dynamic Fisher Personalization (DFP) and Adaptive Constraint (AC), that utilize layer-wise Fisher information to dynamically personalize and adaptively constrain parameters, effectively addressing the aforementioned issues. The efficacy of our proposed methods has been thoroughly validated against many popular counterparts across various classification tasks. We are hopeful that this work will pave the way for future research on personalized federated learning with differential privacy.

**Acknowledgment.** This work is partially supported by National Natural Science Foundation of China under Grant (62176188), the Key Research and Development Program of Hubei Province (2021BAA187, 2022BCA009), Zhejiang lab (NO.2022NF0AB01), CCF-Huawei Populus Grove Fund (CCF-HuaweiTC2022003), and the Special Fund of Hubei Luojia Laboratory (220100015).

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
