# OpenReview forum: "Dynamic Personalized Federated Learning with Adaptive Differential Privacy"
_NeurIPS.cc/2023/Conference — NeurIPS 2023 poster_

### Official Review · Reviewer_UFEt · 2023-06-29

**Soundness:** 4 excellent
**Presentation:** 4 excellent
**Contribution:** 4 excellent
**Rating:** 8
**Confidence:** 4

**Summary:**

This paper addresses the pressing issues of inflexibility and convergence difficulties in personalized federated learning under the influence of differential privacy constraints. The authors propose an innovative adaptive approach that prioritizes high fisher value local parameters during the personalization process while utilizing differential constraint techniques to enhance convergence. Rigorous empirical testing conducted on CIFAR-10 and EMNIST datasets corroborates the effectiveness of the proposed method.

**Strengths:**

- The authors tackle an important problem in the realm of personalized federated learning. Given that current methodologies often rely heavily on fixed personalization elements, the ability to dynamically select parameters for personalization, as proposed in this paper, could potentially enrich the field.

- The addressed problem also matters in the context of model training under differential privacy. While differential privacy can offer stricter privacy protection during model training, its inclusion can significantly interfere with the learning process. The proposed adaptive constraint method, by introducing an adaptive algorithm for differential privacy techniques, effectively addresses this critical issue.

- The proposed technique of incorporating Fisher information into the personalization process is a novel approach. They select parameters based on Fisher information values, which introduces an innovative angle to the personalization process.

- The experimentation in the paper is thorough, considering various influential factors. By conducting experiments under different settings on the CIFAR-10 and EMNIST datasets - such as varying degrees of non-iid data among clients and different privacy budget epsilon settings - the authors demonstrate the effectiveness of the proposed method.

- The paper is clearly written and well-structured. The authors clearly split the problem they are solving into two parts, providing a clear and logical flow that facilitates understanding.



**Weaknesses:**

- Although the authors touch upon some limitations of their proposed methodology, a more comprehensive discussion on potential adverse societal impacts would have strengthened the paper.



**Questions:**

Could the authors  provide further insights into the motivation behind preserving parameters with high fisher values for personalization? How do these parameters enhance the personalization process in the federated learning context?

**Limitations:**

The authors' discussion about potential adverse social impact is sparse. Inclusion of this aspect would enhance the overall analysis.

---

> ### Author Rebuttal · Authors · 2023-08-09
>
> Dear Reviewer UEFt:
>
> Thank you very much for your affirmation of our work, as well as the insightful concerns and questions you have raised. We have carefully considered each comment and provided responses.
>
> **Weaknesses:**
>
> **W1: More discussion on potential adverse social impact**
>
> **A1:** Thank you for pointing this out, we have added the following discussion:
>
> Towards the proposed personalized federated learning approach, if personalization is not handled carefully, it might introduce biases that favor particular groups, which is an important adverse social impact.
>
> Regarding we propose a method that adapts to differential privacy, achieving good generalization while protecting user data privacy. In practical applications, care should be taken to strike a balance between privacy and utility, otherwise, it may lead to adverse social impacts such as inefficient use of data or data leakage.
>
> We will incorporate this discussion into the final version if accepted.
>
> **Questions:**
>
> **Q1: Motivation for personalization strategy and working manner of our personalized approach**
>
> **A1:** The motivation behind preserving parameters with high Fisher values for personalization in our work stems from their ability to quantify the amount of information that these parameters can provide. More informative parameters represent knowledge more effectively and play a crucial role in model prediction. By employing layer-wise Fisher information to measure the information content of client parameters, we dynamically determine the personalized parameter portion for each client, ensuring that these informative parameters remain unaffected by noise introduced for privacy protection.
>
> In the federated learning context, this approach enhances the adaptability of the model to individual clients' data distributions, effectively overcoming the challenges associated with inflexible personalization and the impact of noise, leading to improved personalized performance.

---

> > ### Comment · Reviewer_UFEt · 2023-08-21
> >
> > Thanks for the rebuttal. After reading the authors’ rebuttal and the other reviewers' comments, I think the authors have well addressed the concerns. Missing experiments on hyper-parameters are added. This paper presents a simple yet effective approach for personalized federated learning with adaptive differential privacy. The way of using Fisher information is reasonable and interesting. Moreover, incorporating the adaptive personalization property to differential privacy is instructive. Thus, I keep my original score and vote for clear acceptance of this paper.

---

### Official Review · Reviewer_aCpX · 2023-07-05

**Soundness:** 3 good
**Presentation:** 4 excellent
**Contribution:** 3 good
**Rating:** 8
**Confidence:** 5

**Summary:**

This study generally explores the problem of personalized federated learning with differential privacy. Specifically, the paper proposes two novel methods known as DFP and AC for enhancing the learning capability of a model while reducing the impact of noise, and improving robustness against clipping. At its core, the two methods encompass: 1) An adaptive scheme that selects parameters characterized by high Fisher values for personalization parts, and 2) different constraints on parameters, based on the information they carry. The experiments were conducted across multiple tasks and settings and have shown promising results, demonstrating the efficacy of the proposed method.

**Strengths:**

1. This study addresses a critical challenge in the field of federated learning and privacy preservation, particularly in the context of personalized federated learning with differential privacy. The proposed methods show promise in mitigating associated challenges, and offer a reasonable balance between model personalization, privacy protection, and the impact of noise, presenting a noteworthy contribution to the federated learning domain.

2. The authors propose modifications to the client's local training procedure that could potentially enhance the robustness of the training results against differential privacy operations. This is an insightful contribution that seems to manage the trade-off between privacy and utility in federated learning, an important consideration in environments where privacy is paramount.

3. The paper introduces the use of Fisher information values in the personalization process, which is a novel approach within the realm of federated learning. This offers a dynamic identification of personalized parts in each round, providing more flexibility compared to the current fixed portion personalization methods.

4. The literature review in Section 2 of this paper is comprehensive. It provides a thorough overview of federated learning methods under differential privacy, all of which appear to be current and relevant.

5. The experimental results in the study are promising. The proposed methods appear to perform better than the state-of-the-art methods under various settings. This, combined with the in-depth experimental section, suggests the potential robustness and efficacy of the proposed methods in personalized federated learning with differential privacy.


**Weaknesses:**

1. The analysis in Section 3.3 is not thorough enough when it comes to measuring complexity. The authors only focus on the number of parameters P and overlook important training factors like the size of the dataset.

**Questions:**

1. I'm curious if the DFP technique still maintains its effectiveness when differential privacy constraints are absent. Could the authors elaborate on this and explain the underlying reasons?

2. Considering the paper's emphasis on differential privacy for client-side protection, it provokes my consideration of the proposed method's scalability and interoperability with other privacy-preserving techniques. Could it be possible for the introduced approach could be seamlessly integrated with other strategies, such as secure aggregation, for enhanced privacy protection?

**Limitations:**

The discussion on limitations appears narrow, predominantly focusing on model heterogeneity. It could benefit from discussing further limitations and potential negative societal impact.

---

> ### Author Rebuttal · Authors · 2023-08-09
>
> Dear Reviewer aCpX:
> Thank you very much for your recognition of our work and for raising thoughtful questions and concerns about our work. We have carefully considered each comment and provided responses.
>
> **Weaknesses:**
>
> **W1: Complete complexity analysis**
>
> **A1:** Our analysis of extra computation cost is as follows:
>
> Assuming a total of $N$ Global Epochs, $M$ clients, and $P$ parameters for each model, during each client update process, the client undergoes $S$ steps of parameter updates. In each round, our method introduces the following extra computation costs:
>
> - Computing the Fisher information value
> - Construction of the mask
> - Calculating the starting parameters based on the above results
>
> In these three steps, considering that each step is parameter-specific, our method brings an additional computational cost of $\mathcal{O}(3 \cdot P)$, regarding that we have to carry out these operations once in each global epoch, we introduce an extra computational cost of $\mathcal{O}(3 \cdot E \cdot M \cdot N \cdot P)$ over the entire training process.
>
> Despite our extra computational cost, it is noteworthy that our approach has a relatively **small overhead** compared to the overall training process. This is because, in the entire update process, the model's training still has a computational cost proportional to $P$, and considering all the training process would introduce $\mathcal{O}( E \cdot M \cdot N \cdot P \cdot S)$. Generally, in a training process, the value of $S$ is in the tens or hundreds, so the computation cost introduced by our method will be an order of magnitude or more less than the training expense. Extensive experiments have verified the consistent improvement over the counterparts.
>
> We will include this analysis into the final version of our work if accepted.
>
> **Questions:**
>
> **Q1: Effectiveness of our method under other conditions**
>
> **A1:** Our Dynamic Fisher Personalization (DFP) technique will retain its effectiveness even in a scenario without differential privacy protection. The effectiveness of this Personalization lies in retaining parts that have learned client-specific key knowledge within the client during the personalization process, which can simultaneously enhance the personalized effect while reducing the impact of noise on model training.
>
> **Q2: Feasibility of integration with other methods**
>
> **A2:** Considering that our method does not have any operations conflicting with encryption during the process of sending the gradient back to the server, it is feasible to integrate secure aggregation methods into our framework, demonstrating the scalability of our approach.

---

> > ### Comment · Reviewer_aCpX · 2023-08-17
> > **Responses to Authors' Rebuttal**
> >
> > I have viewed the authors' rebuttals for my questions and others. I appreciate this manuscript with respect to the targeted task, the proposed method and the experimental results, and beyond this, some concerns and questions about this manuscript have been positively responded. To sum up, I keep my score.

---

### Official Review · Reviewer_EfaD · 2023-07-05

**Soundness:** 3 good
**Presentation:** 3 good
**Contribution:** 2 fair
**Rating:** 5
**Confidence:** 4

**Summary:**

This paper proposes a dynamic personalization strategy using layer-wise Fisher information values, which enables flexible personalization and reduced noise impact. This paper further presents an adaptive method for enhancing clipping robustness by adjusting constraint strengths for different parameters, balancing client-specific features and global knowledge while minimizing clipping effects.

**Strengths:**

1. This paper is genrally well written.
2. Proposed method seems novel.
3. Experimental results under multiple tasks and settings seem promising.

**Weaknesses:**

1. Experiments are limited to simple datasets CIFAR-10 and FEMNIST datasets.
2. Analysis on extra computation cost is expected. As mentioned in Sec 3.3, seems adding a computational overhead proportional to the number of parameters is huge, especially for deep models?
3. For CIFAR-10, this paper evaluates average accuracy under different degrees of non-IID data partition, which seems unusual. What about the performance for different degrees of non-IIDness?

**Questions:**

1. Analysis on extra computation cost is expected. As mentioned in Sec 3.3, seems adding a computational overhead proportional to the number of parameters is huge, especially for deep models?
2. For CIFAR-10, this paper evaluates average accuracy under different degrees of non-IID data partition, which seems unusual. What about the performance for different degrees of non-IIDness?
3. What "robust privacy" means? seems robustness and privacy are two seperate topics?

**Limitations:**

yes

---

> ### Author Rebuttal · Authors · 2023-08-09
>
> Dear Reviewer EfaD:
>
> Thank you for your thoughtful review and for raising key concerns regarding our work. We aim to address your concerns in our detailed responses below, hoping to provide clarity and demonstrate the effectiveness of our proposed approach.
>
> **Weaknesses:**
>
> **W1: Experiments are limited to simple datasets**
>
> **A1:** To enhance the credibility and robustness of our results, we have conducted experiments on the SVHN dataset, which is a larger and more complex dataset. The experiment results are as follows:
>
> Table: Comparison of model performance under different epsilons
>
> | Methods|ε = 2|ε = 4|ε = 6|ε = 8|
> |-|-|-|-|-|
> | DP-FedAvg|53.91|54.76|56.33|57.65|
> | BLUR+LUS|53.76|57.52|56.90|58.57|
> | PPSGD |55.89|56.15|56.74|59.37|
> | DP-FedSAM|53.04|52.84|54.03|56.08|
> | Ours |58.78|62.63|63.66|64.57|
>
> Experiments demonstrate that our method still surpasses other state-of-the-art approaches in a larger and more complex dataset, showcasing the effectiveness of our method in different scenarios.
>
> We will incorporate this experiment into the final version if accepted.
>
> **W2: Analysis of extra computation cost**
>
> **A2:** Our analysis of extra computation cost is as follows:
>
> Assuming a total of N Global Epochs, M clients, and P parameters for each model, during each client update process, the client undergoes S steps of parameter updates. In each round, our method introduces the following extra computation costs:
>
> 1. Computing the Fisher information value
>
> 2. Construction of the mask
>
> 3. Calculating the starting parameters based on the above results
>
> In these three steps, considering that each step is parameter-specific, our method brings an additional computational cost of $\mathcal{O}(3 \cdot P)$, regarding that we have to carry out these operations once in each global epoch, we introduce an extra computational cost of $\mathcal{O}(3 \cdot E \cdot M \cdot N \cdot P)$ over the entire training process.
>
> Despite our extra computational cost, it is noteworthy that our approach has a  **relatively small** overhead compared to the overall training process. This is because, in the entire update process, the model's training still has a computational cost proportional to P, and considering all the training process would introduce $\mathcal{O}( E \cdot M \cdot N \cdot P \cdot S)$. Generally, in a training process, the value of S is in the tens or hundreds, so the computation cost introduced by our method will be **an order of magnitude** or more less than the training expense. Extensive experiments have verified the consistent improvement over the counterparts.
>
> **W3: Performance under different degrees of non-IIDness**
>
> **A3:** In our paper, we explore the performance under varying degrees of non-IIDness, as presented in Table 1, as the degree of heterogeneity escalates ($\alpha$ changing from 1 to 100, with IID representing infinite $\alpha$), the model performance experiences degradation. This observation underscores that heightened heterogeneity can compromise both robustness and performance.  However, compared with other baselines, the performance of degradation on our method is minor, demonstrating the robustness of our method, as verified in the following experiment
>
> | Method    | α = 0.1 | α = 0.3 | α = 0.5 |
> |-----------|---------|---------|---------|
> | DP-FedAvg | 24.65   | 30.07   | 38.43   |
> | BLUR+LUS  | 23.98   | 33.20   | 40.67   |
> | PPSGD     | 25.04   | 36.48   | 39.42   |
> | DP-FedSAM | 21.20   | 28.11   | 34.05   |
> | Ours      | 25.17   | 37.66   | 41.34   |
>
> **Questions:**
>
> **Q1: Analysis of extra computation cost**
>
> **A1:** As replied above.
>
> **Q2: Performance under different degrees of non-IIDness**
>
> **A2:** As replied above.
>
> **Q3: Meaning of “robust privacy”**
>
> **A3:** We use the "robust privacy in Section 2.2 of our manuscript, we intended to emphasize the enhanced and strict privacy assurance offered by the Differential Privacy mechanism. This notion signifies a more fortified level of privacy protection compared to the conventional "data-stay-local" privacy concept associated with Federated Learning.
>
> We will improve the clarity to avoid misleading understanding in the final version.

---

### Official Review · Reviewer_imA4 · 2023-07-05

**Soundness:** 3 good
**Presentation:** 2 fair
**Contribution:** 2 fair
**Rating:** 5
**Confidence:** 4

**Summary:**

This paper proposes an adaptive method for Personalized Federated Learning with Differential Privacy (DP-PFL), which aims to address the inflexible personalization and convergence difficulties that DP-PFL suffers from. Firstly, layer-wise Fisher information is used to measure the information content of local parameters. Parameters with high Fisher values are retained locally and prevented from noise perturbation. Secondly, an adaptive approach by applying differential constraint strategies to personalized parameters and shared parameters identified in the previous step is proposed for better convergence. Experimental results on CIFAR-10 and FEMNIST datasets demonstrate the effectiveness of the approach in achieving better personalization performance and robustness against clipping.

**Strengths:**

S1. A dynamic personalization strategy using layer-wise Fisher information values is proposed, enabling flexible personalization and reducing the impact of noise.

S2. This paper presents an adaptive method for enhancing clipping robustness by adjusting constraint strengths for different parameters.

S3. Extensive experiments under multiple tasks and settings, including an ablation study, are conducted to demonstrate the effectiveness and robustness of the approach.


**Weaknesses:**

W1. The paper lacks technical novelty. For example, this work mainly borrowed the idea from statistical estimation theory as a measurement to represent the amount of information content, and the other part of the proposed solution is straightforward.

W2. The paper is sometimes self-conflicting. For example, in the introduction at Line 62, it is stated that the second-order derivative of the log-likelihood function is computed to get the Fisher value. However, in Equation 4, a square of the first-order derivative is computed instead, without adequate explanation.

W3. The experiments are insufficient, and more experiments are suggested to demonstrate how the proportion of shared parameters changes with different settings of hyperparameters. Moreover, it is critical to show how the threshold affects the accuracy of the method.

W4. There are some presentation issues and typos. For example, in Line 356, “…via our flexible personzalzation method” should be “…via our flexible personalization method”.


**Questions:**

Except for the weak points, I have the following questions:

Q1. Can you please provide more details on the mechanism of the approach, particularly how you modify the Fisher Information Matrix in statistical estimation theory to adapt to DP-PFL, and how it looks originally?

Q2. In Section 3.1, the definition of Personalized Federated Learning is not described well, and the variables are not precisely expressed in the process of Local Update and Collaborative Update. Please review this section carefully and make the necessary corrections.

Q3. The setting of epsilon is a little large, which could be impractical for real-world applications. Please provide more justifications.

---

> ### Author Rebuttal · Authors · 2023-08-09
>
> Dear Reviewer imA4:
>
> Thanks for raising thoughtful questions and concerns. We sincerely hope this point-to-point response allows the reviewer to update the score.
>
> **Weaknesses:**
>
> **W1: Lacks technical novelty**
>
> **A1:** We did not merely adopt statistical estimation theory into our work. Rather, we approached it with unique insights and reasoning, tailoring it specifically for our application. Conscious alterations and improvements were implemented to ensure its effectiveness in our scenario.
>
> For Personalization in FL, our novelty does not lie in the utilization of Fisher information. Existing personalization methods often depend on FIXED techniques by manually selecting personalized parts, which are inflexible under different data distributions. We provide a new angle to evaluate the information importance of parameters in the personalization process, which can adaptively select personalization parameters with a slight additional cost. Rather than simply repeating the Fisher calculation in traditional information field, we aligned it with the traits in deep neural networks, introducing a layer-wise Fisher calculation. This provides a more accurate selection of personalized parameters, finely tuned to the specific requirements of each layer. We believe this simple yet effective approach provides insights for the community.
>
> In addition, we also design an adaptive Differential Privacy strategy, our novelty is reflected in designing different constraint strategies for parameters to make gradient adapt to the clipping operations required by differential privacy. Previous gradient computation methods are unsuitable for clipping operations, leading to difficulties in model convergence. We propose a new strategy that enables the gradient to adapt to the clipping operation, contributing to better convergence. Considering that Reviewer dw3B, EfaD, aCpX, and UFEt have all agreed that our method is a novel approach and our proposed method outperforms other state-of-the-art methods in experiments, we believe our research findings are worth sharing with the community.
>
> **W2: Equation 4 conflicts with the Introduction**
>
> **A2:** We are sorry for the misleading presentation. The fisher information value is actually defined as the square of the first-order derivative of the log-likelihood function as Eq.4 demonstrates.
> To prevent misunderstandings, we will correct the relevant part of the Introduction in the final version if accepted.
>
>
> **W3: Additional experiments on $\tau$**
>
> **A3:** We have extended our experiments as follows:
>
> Table: Ablation study on $\tau$
> |$\tau$|0|0.1|0.2|0.3|0.4|0.5|0.6|0.7|0.8|0.9|1|
> |-|-|-|-|-|-|-|-|-|-|-|-|
> |Proportion|100%|11.1%|4.5%|1.45%|0.97%|0.46%|0.15%|0.1%|0.1%|0.05%|0%|
> |Accuracy|39.72|47.97|49.28|48.89|48.35|48.03|48.28|48.14|48.35|47.74|48.29|
>
> Proportion is the proportion of personalization parameters. Experimental results show that the proportion of personalized parameters decreases as \tau increases, and our method can achieve the best results when the value of \tau is around 0.2. This demonstrates the effectiveness and indispensability of our personalization approach.
>
> Generally, our proposed method is relatively robust against different $\tau$s, verifying the effectiveness and indispensability of our personalization approach. We will incorporate these experiments into the final version if accepted.
>
> **W4: Presentation issues and typos**
>
> **A4:** We have corrected the error you noted on Line 356 and other presentation issues.
>
> **Questions:**
>
> **Q1: Rationale of mechanism and Fisher Information Matrix**
>
> **A1:** The original form of the Fisher Information Matrix is defined as:
> $$
> F=\frac{1}{N} \sum_{i=1}^N \nabla \log p(x_i | \theta) \nabla \log p(x_i | \theta)^T,
> $$
> Considering the values outside the main diagonal of the matrix reflect the mixed information content between different parameters, which is not our concern, and the elements on the main diagonal uniquely correspond to the parameters, reflecting the sensitivity of the log-likelihood function of each parameter, we directly compute the values in the main diagonal to represent the information content corresponding to each parameter, as shown in Eq.4.
> In the process of adapting to our algorithm, we did not simply apply this computation method to neural networks. Instead, considering traits in the neural network that distribution of Fisher Values of parameters in different layers is naturally different, we introduced the layer-wise Fisher Value, leading to a more precise personalization.
>
> **Q2: Revise Section 3.1 for clarity and precision**
>
> **A2:** We will make the following changes to address your comments:
> - We will improve the description of each client private dataset by explicitly stating the range of i as $i=1,..., N_m$.
> - We will redefine the "Local Update" part, it will delineate that each client adopts the latest global parameter $v^t$ while maintaining their local parameters $u_i^t$ at the beginning of local update round.
> - In the "Collaborative Update" part, we will stress that all clients transmit their global parameter updates $\Delta v^{t+1}$ to the server.
>
> We will incorporate these changes into the final version if accepted.
>
> **Q3: Justifications for a little large ε value**
>
> **A3:** Our initial selection of epsilon in (2, 4, 8, 16) was intentional, by selecting these exponentially growing values, we aim to showcase how well our method works across a broad range of ε. We also conduct additional experiments on SVHN dataset with epsilon in (2, 4, 6, 8). The results are in the table below:
>
> |Methods|ε = 2|ε = 4|ε = 6|ε = 8|
> |-|-|-|-|-|
> |DP-FedAvg|53.91|54.76|56.33|57.65|
> |BLUR+LUS|53.76|57.52|56.90|58.57|
> |PPSGD|55.89|56.15|56.74|59.37|
> |DP-FedSAM|53.04|52.84|54.03|56.08|
> |Ours|58.78|62.63|63.66|64.57|
>
> Experiments demonstrate that even under the setting of smaller epsilon, our method still surpasses other state-of-the-art approaches, showcasing the effectiveness of our method.

---

> > ### Comment · Reviewer_imA4 · 2023-08-16
> > **Response to the author rebuttal**
> >
> > In the rebuttal, the authors have responded all my weak points and questions. Overall, some of my comments were well explained, such as W2 and W3; some of them were planned to be revised during the camera ready (if the paper is accepted), such as Q2.
> >
> > One thing, which I expected more, is Q3. My question is to **justify the rationale of selecting proper epsilon with real-world applications** instead of **conducting more experiments**. In other words, why do you think epsilon=2 has already been rigorous enough to protect privacy in this paper's application scopes? Perhaps the authors might want to explain it with experiments. Even If that's the intention, the authors should try much smaller epsilon (i.e., more rigorous privacy protection), like 0.1.
> >
> > After a thoughtful consideration, I raised my score, since most of my major concerns were explained.

---

> > > ### Author Response · Authors · 2023-08-16
> > >
> > > Dear Reviewer imA4:
> > >
> > > Thank you very much for recognizing our rebuttal and raising the score. In response to your Q3, we have conducted further experiments and analyses accroding to your suggestions, detailed as follows:
> > >
> > > The selection of $ \epsilon $ in real-world applications is a trade-off between privacy and the utility of the model. When $ \epsilon $ is set too small, it means that adding too much noise to the network, the training process may become overwhelmed with noise, causing the model to fail to converge, severely affecting its utility. Conversely, if $ \epsilon $ is too large, it introduces minimal noise, potentially leading to privacy concerns.  To verify this point, we further extend the experiments in Q3 for $ \epsilon $, as shown in the following Table.
> > >
> > > | Methods   | ε = 0.1 | ε = 0.5 | ε = 1   | ε = 2  | ε = 4  | ε = 6  | ε = 8  |
> > > |-----------|--------|--------|--------|-------|-------|-------|-------|
> > > | DP-FedAvg | - (19.58) | - (19.56) | 45.22 | 53.91 | 54.76 | 56.33 | 57.65 |
> > > | BLUR+LUS  | - (19.22) | - (18.99) | 45.84 | 53.76 | 57.52 | 56.90 | 58.57 |
> > > | PPSGD     | - (19.58) | 44.81     | 47.86 | 55.89 | 56.15 | 56.74 | 59.37 |
> > > | DP-FedSAM | - (11.07) | 23.86     | 47.03 | 53.04 | 52.84 | 54.03 | 56.08 |
> > > | Ours      | 39.48    | 50.34     | 57.29 | 58.78 | 62.63 | 63.66 | 64.57 |
> > >
> > > In the table, entries marked with "-" indicate models failed to converge.
> > >
> > > From the experiments, we observe that under the rigorous privacy setting of $ \epsilon = 0.1 $, most other models failed to converge, and for $ \epsilon $ values of 0.5 and 1, all models exhibited a significant drop in accuracy. Demonstrating that a too small $ \epsilon $ setting sacrifices too much performance to ensure privacy.
> > >
> > > Notably, our method is more robust to changes in the privacy budget $\epsilon$ compared to other methods. This is attributed to our dynamic personalization, which effectively reduces the increased noise interference caused by a smaller privacy budget. Moreover, our method achieves good results under different settings of $\epsilon$, demonstrating its effectiveness. In practice, based on our experience and extensive experiments, we believe that an $\epsilon$ value between 2 and 8 strikes a good balance and can be effectively applied in real-world scenarios. This range effectively safeguards user privacy while preventing the model from collapsing during training due to overly stringent privacy settings, thus realizing a good trade-off between privacy and utility. We will also further explore more reasonable choices of epsilon values in our future work
> > >
> > > We hope our response addresses your concerns.

---

> > > > ### Comment · Reviewer_imA4 · 2023-08-17
> > > > **Response to the rebuttal**
> > > >
> > > > Thanks for the prompt response to my Q3.

---

### Official Review · Reviewer_dw3B · 2023-07-27

**Soundness:** 2 fair
**Presentation:** 3 good
**Contribution:** 2 fair
**Rating:** 5
**Confidence:** 1

**Summary:**

This paper proposes a novel method for Dynamic Personalized Federated Learning with Adaptive Differential Privacy. The method addresses the challenges of inflexible personalization and convergence difficulties in current DP-PFL methods by leveraging layer-wise Fisher information and introducing an adaptive approach. The proposed method retains informative local parameters and prevents noise perturbation, resulting in better personalization performance and robustness against clipping. The experiments on CIFAR-10 and FEMNIST datasets demonstrate the effectiveness of the proposed method in achieving better personalization performance and robustness against clipping under differential privacy personalized federated learning.

**Strengths:**

- The paper is well-written, with clear organization.
- The research questions are well-defined and relevant, backed by a comprehensive literature review.
- The paper presents compelling results, demonstrating the effectiveness of the proposed approach
compared to state-of-the-art techniques in small-scale experiments.

**Weaknesses:**

- The way in which adaptive differential privacy is defined lacks clear motivation and appears to be a composition
of not well-motivated components. The reasons for introducing losses in equations (9) and (8),
as well as how these losses are combined to obtain the adaptive constraint in section 3.2.2, are not clearly elucidated.
- The paper would benefit from running experiments in a larger context. While the results in small size benchmarks are promising, their applicability to real-world scenarios needs to be validated on more extensive datasets to observe the model's effectiveness at scale.
- An ablation study on hyperparameters \lambda and \tau is necessary. Understanding how the model adapts or resists new parameter updates and how the two masks are computed based on the Fisher Information Matrix requires thorough investigation. The absence of such a study leaves questions about the robustness and sensitivity of the proposed approach unanswered.

minors:
- In Eq. 4, consider increasing the size of the brackets for better readability.
- In Eq. 7, there are different symbols used for the global and local parameters of the network, u and v appear to be used inconsistently. This somewhat reduces clarity.

**Questions:**

- Are the values of \tau and \lambda dataset-dependent, or are they independent of the specific datasets used in the experiments?
- In scenarios where all local users have different datasets, how does this heterogeneity affect the performance and robustness of the federated learning approach proposed in the paper?
- In Table 2, why does the combination of DFP and AC yield the highest values for \epsilon = 8, whereas the highest performance of DFP and AC alone is achieved for \epsilon = 16? What factors contribute to this discrepancy in performance for different privacy budgets?

**Limitations:**

Limitations are addressed. Explicit mention of ethical aspects are lacking, however they can be implicitly included in federated learning.

---

> ### Author Rebuttal · Authors · 2023-08-09
>
> Dear Reviewer dw3B:
>
> Thanks for your insightful comments and questions. We sincerely hope our detailed response allows the reviewer to update the score.
>
> **Weaknesses:**
>
> **W1: Motivation of adaptive differential privacy and equations 8 and 9, Details about Section 3.2.2**
>
> **A1:** Current Differential Privacy operations require clipping the gradients in their magnitude. However, existing methods only focus on minimizing empirical risk and cannot handle the varying and inconsistent gradient norm, as we elaborate in our main manuscript (Line 51 in Page 2). As a result, they exhibit poor robustness to norm clipping, and the distortion caused by the clipping of the gradient can affect the model's convergence.  To address this issue, we develop an adaptive solution under the personalized framework.
>
> Considering that the parameter 'u' represents vital knowledge learned from the client dataset and reflects its specific characteristics, this knowledge is indispensable in personalized federated learning. To retain this knowledge, consequently, we introduce a loss function that restricts the update size of the informative local parameter 'u' by adding a regularization term and also helps in mitigating the impact of clipping and achieving adaptation to differential privacy, as shown in equation (9). Equation (8) represents the process of updating the parameter 'u' using equation (9).
>
> Finally, the adaptive constraint is obtained by updating the 'u' parts of the parameters using equation (8) and updating the 'v' parts of the parameters using equation (11), as detailed in 8-Algorithm 1. Extensive experiments have verified this simple yet effective operation can consistently improve the performance under various settings.
>
> **W2: Additional real-world scenarios experiments**
>
> **A2:** We have expanded our experiments to include the Street View House Numbers (SVHN) dataset, which originates from real-world street view scenarios.
>
> Table: Comparison of model performance under different epsilons
>
> | Methods|ε = 2|ε = 4|ε = 6|ε = 8|
> |-|-|-|-|-|
> | DP-FedAvg|53.91|54.76|56.33|57.65|
> | BLUR+LUS|53.76|57.52|56.90|58.57|
> | PPSGD |55.89|56.15|56.74|59.37|
> | DP-FedSAM|53.04|52.84|54.03|56.08|
> | Ours |58.78|62.63|63.66|64.57|
>
> Experiments demonstrate that our method still surpasses other state-of-the-art approaches in a larger and more complex dataset from real-world scenarios, showcasing the effectiveness of our method in different scenarios.
> We will include this experiment in the final version if accepted.
>
> **W3: Ablation study on hyperparameters $\lambda$ and $\tau$**
>
> **A3:** We have conducted the ablation experiments, and the results are presented in the following table:
>
> Table: Ablation study on $\tau$
>
> |$\tau$|0|0.1|0.2|0.3|0.4|0.5|0.6|0.7|0.8|0.9|1|
> |-|-|-|-|-|-|-|-|-|-|-|-|
> |Proportion|100%|11.1%|4.5%|1.45%|0.97%|0.46%|0.15%|0.1%|0.1%|0.05%|0%|
> |Accuracy|39.72|47.97|49.28|48.89|48.35|48.03|48.28|48.14|48.35|47.74|48.29|
>
> For $\tau$, we observe that our proposed method is relatively stable, verifying the robustness of our proposed solution.
>
> Table: Ablation study on $\lambda_1$
>
> |$\lambda_1$|0|0.05|0.1|0.3|0.5|
> |-|-|-|-|-|-|
> |Accuracy|60.82|62.64|59.66|59.28|59.57|
>
> Table: Ablation study on $\lambda_2$
>
> |$\lambda_2$|0|0.05|0.1|0.3|0.5|
> |-|-|-|-|-|-|
> |Accuracy|58.34|59.76|61.01|47.61|45.66|
>
> For $\lambda$, we achieve the best performance when $\lambda1$ is around 0.05 and $\lambda2$ is around 0.1. We apply this selection across multiple different datasets, and it results in a consistent good performance. In the future, we will explore a more flexible solution for the selection of $\lambda$. We will incorporate these experiments with above analysis into the final version if accepted
>
> **W4: Revise Eq. 4, Eq. 7**
>
> **A4**: Thanks for pointing out these issues. We will revise Eq. 4 and Eq. 7 to ensure clarity and consistency.
>
> **Questions:**
>
> **Q1: Whether the $\tau$ and $\lambda$ depends on the dataset**
>
> **A1:** Please refer to the ablation study table of $\tau$, our proposed method is relatively stable when the value of $\tau$ changes. Based on what our experiments have shown, generally speaking, a value of $\tau$ around 0.2 can achieve good personalized effects, and continuing to expand its value does not cause serious performance degradation, verifying the robustness of our proposed solution.
>
> For $\lambda$, we find that choosing $\lambda1$ to be around 0.05 and $\lambda2$ to be around 0.1 across multiple datasets results in consistent good performance. This reflects the consistency of the hyperparameters across different datasets, demonstrating the effectiveness of our method.
>
> **Q2: How heterogeneity affects performance and robustness**
>
> **A2:** As data heterogeneity rises, the local data distributions become more biased towards the global distribution, resulting in more chaotic local updates. Consequently, the clipped local updates tend to align less with the original local updates, adding complexity to convergence.
>
> In the experiments, we specifically explore the performance under varying degrees of non-IIDness, as presented in Table 1, as the degree of heterogeneity escalates ($\alpha$ changing from 1 to 100, with IID representing infinite $\alpha$), the model performance experiences degradation. This observation underscores that heightened heterogeneity can compromise both robustness and performance. However, despite the degradation in performance, compared with other baselines, the degradation on our method is minor, demonstrating the robustness of our method.
>
> **Q3: Reason why not having the highest score when $\epsilon = 16$**
>
> **A3:** We believe that the observed variation in performance is due to statistical fluctuations that can occasionally occur in experiments. In order to address this, we repeated the experiments and obtained accuracy scores of 77.59%, 77.40%, and 78.31%, with an average of 77.76%. This average score represents the highest accuracy across all epsilon settings.

---

> > ### Author Response · Authors · 2023-08-20
> >
> > Dear Reviewer dw3B:
> >
> > We extend our sincere gratitude to the reviewer for their valuable time and insightful feedback. We value the constructive feedback and hope that our response has appropriately addressed all the concerns. Further, we are happy to address any remaining concerns.

---

### Decision · Program_Chairs · 2023-09-21

**Decision:**

Accept (poster)

**Comment:**

A new method is presented to address the challenges of personalization and privacy in FL by retaining informative local parameters while reducing the impact of noise, resulting in better personalization and robustness against clipping. Experiments are run on  CIFAR-10 and FEMNIST datasets and show the effectiveness of the proposed method.  Reviews are fairly consistent and positive.